# Assessment of Resistance of Different Varieties of Winter Wheat to Leaf Fungal Diseases in Organic Farming

Paweł Radzikowski [1,*], Krzysztof Jończyk [1], Beata Feledyn-Szewczyk [1] and Tomasz Jóźwicki [2]

[1] Department of Systems and Economics of Crop Production, Institute of Soil Science and Plant Cultivation—State Research Institute in Puławy, 24-100 Puławy, Poland
[2] Department of Agrometeorology and Applied Informatics, Institute of Soil Science and Plant Cultivation—State Research Institute in Puławy, 24-100 Puławy, Poland
* Correspondence: pradzikowski@iung.pulawy.pl; Tel.: + 48-81-4786-820

**Abstract:** A change in agricultural policy in the European Union aims, among other things, to halve the use of pesticides and increase the share of organic farming to 25% by 2030. One of the challenges associated with this target will be the control of plant fungal diseases. The key methods in organic farming include the selection of less susceptible crop varieties. In order to test this method, a long-term trial of organic farming in Eastern Poland was established. In total, 41 different winter wheat varieties were grown from 2018 to 2022 and their resistance to fungal leaf diseases was monitored. Brown rust was found to be the disease causing the highest infestation towards the end of vegetation, often exceeding 80% of the flag leaf area. However, yield reductions were mainly related to the severe occurrence of leaf Septoria. Other leaf diseases such as tan spot, yellow rust, powdery mildew, and fusariosis were of little importance and only occurred at low infestations of Septoria and brown rust. The course of the weather was found to have a significant effect on disease incidence. Drought occurring in May and June significantly increased the incidence of brown rust and Septoria, while prolonged rains increased tan spot and Fusarium infestation. Greater overall infestation occurred in years with high average temperatures. Ten varieties with high resistance to foliar fungal diseases were selected and can be recommended for organic farming.

**Keywords:** organic farming; fungal diseases; winter wheat

## 1. Introduction

Fungal diseases of winter wheat are one of the most yield-limiting environmental factors. On average, they reduce grain yield by 20%; however, in some years losses can exceed 50%. Fungal infections are responsible for half of all cereal diseases [1]. They also affect the quality of grain and case contamination of food products with dangerous mycotoxins. The most important fungal diseases of cereals in temperate climates are powdery mildew *Blumeria graminis* (DC.) Speer, rusts *Puccina recondite* Roberge ex Desm., *Puccinia graminis* Pers., *Puccinia striiformis* Westend., Septoria *Mycosphaerella graminicola* (Fuckel) J. Schröt., *Phaeosphaeria nodorum* (E. Müll.) Hedjar., tan spot *Pyrenophora tritici-repentis* (Died.) Drechsler, and fusariosis *Fusarium* sp. In the last two decades, there have been spiking problems with fungal diseases of cereals worldwide [2,3]. This is caused by various factors related to changes in modern agriculture. In Poland, as in the whole of Europe, livestock production is declining and with it the share of fodder and perennial crops. On the other hand, the share of consumption cereals, especially wheat, is increasing. In Poland, cereal crops account for over 70% of the sowing structure. The specialization of agricultural production contributes to the fact that cereal monocultures are being used more frequently [4,5]. International transport of grain also brings the risk of a higher occurrence of diseases, often far away from its original range. Wheat grain can be transported considerable distances and in large quantities. Many pathogens, including those causing fungal diseases, are transmitted through the grain. However,

it is doubtful that imported wheat will be used for sowing, as seed imports are strictly controlled. Nevertheless, the continuous transportation, transfer, and even the need for disposal provide some opportunity for the spread of plant pathogens [6,7].

The most important factors are weather and climate, which have been changing in the recent century. In Poland, the average annual temperature has increased by one degree in the period of last 50 years alone. Precipitation has also increased by an average of 30 mm per year; however, its distribution is currently quite unfavorable. It is characterized by long periods without precipitation, which, when it does come, is often abrupt and brief. In comparison to the historical observations, more precipitation now occurs during the winter season, when plants cannot take advantage of it. The ratio of temperature to precipitation is unfavorable, and for very many years we have recorded evapotranspiration much higher than precipitation. The result is that in the last two decades, most of the years were characterized by a strongly negative soil water balance [8]. Since the 1970s, groundwater levels have dropped by an average of one meter in the territory of Poland. Drought has become a common phenomenon, occurring almost every year in most of the regions. Lack of access to water causes plants to be significantly weakened, which can make them more susceptible to disease. Despite the provisions and intensive measures taken to protect climate stability, it is almost certain that these phenomena will continue. Fungal diseases can be expected to become an increasing problem in a face of changing climate around the world [1,9,10].

Until recently, the problems of fungal diseases and their prevention were considered separately for organic and conventional agriculture. The latter traditionally relied mainly on chemical plant protection [11,12]. Meanwhile, all solutions developed in organic farming can be used in conventional farming as well. Farmers throughout the European Union will have to learn organic plant protection methods, no matter what cropping system they are using. The European Union's Farm to Fork Strategy aims to halve the use of active pesticide substances per hectare by 2030 [13]. Options in plant protection are becoming limited also due to growing public opposition to pesticide use in food production. In Poland alone, 14 active substances have been withdrawn from use in the last 5 years, significantly limiting their choice as well as the effectiveness of plant protection products. The main reasons for reluctance to use pesticides is growing awareness of consumers about pesticide residues in food and their negative impact on the environment. In the case of fungicides, their impact on beneficial soil mycoflora, including mycorrhizal fungi, has been already proven [14,15].

The EU's New Green Deal sets the goal of increasing the area under organic cultivation to 25% by 2030. This means that huge areas of cultivation will be completely excluded from chemical plant protection [13]. In organic farming, plant diseases are controlled by a wide range of methods. The most obvious solution is to replace synthetic fungicides with natural products, which are allowed in the European Union and individual member states. In the case of cereals, however, the plant protection products allowed in organic farming are seen by Polish farmers as more expensive and less effective than conventional pesticides. This is a completely incorrect assumption, as their effectiveness is sometimes comparable to their conventional counterparts. Nevertheless, agrotechnical methods such as resilient species and variety selection, extended crop rotation, straw removal, or tillage modifications remain the main ways to control the impact of fungal diseases [16–19].

The selection of varieties is a complicated matter in organic farming, as actors with different interests are involved in the process. Often the ideas of organic agriculture do not coincide with advances in plant breeding. Seed producers and selling companies try to meet farmers' expectations by supplying varieties with the highest value of desirable traits, mainly high yield potential, a high seed weight, high quality related to good plant health, environmental resistance, and low cultivation requirements if it is possible. Plant resilience is most easily achieved by transferring desired genes from other related species. However, the use of such varieties is not allowed in organic agriculture, which greatly limits the possibilities of obtaining resistant varieties [20]. In the traditional breeding process, great progress has been made through the identification of genes responsible for resistance to

specific diseases [21]. In the last 30 years, cultivar selection has mainly been directed against powdery mildew, Septoria, tan spot, and to a lesser extent, against rusts [22]. Performance of different winter crop cultivars may differ depending of technology used, especially under specific farming systems. Even the best yielding ones can underperform in nonoptimal conditions. Lack or reduction of basic production measures, such as mineral fertilizers and chemical protection, can bring unexpected results. Therefore, newly registered varieties need to be tested under different environmental conditions, especially for organic farming [23].

The aim of the work presented is (I) to assess the effect of local climatic conditions on the incidence of major fungal diseases and (II) to identify the most resistant winter wheat varieties for organic cultivation while preserving a high degree of diversity. Therefore, the goal was to identify at least five different varieties with no statistical difference in their long-term performance.

## 2. Materials and Methods

### 2.1. Localization

The research was conducted in the eastern part of Poland, in Osiny, N: 51.464349, E: 22.053279. The area is located at the eastern end of the Central European Plain in the Central Poland Plain province, in the South Podlasie Plain macro-region. The area has a humid continental climate (Dfb) according to the Köppen Climate Classification, with mild summers and cool winters with no dry season in the summer. Precipitation occurs throughout the whole year, with the highest amounts recorded in July and August. In the last 2 decades, the average annual temperature was 8.6 °C and the average annual precipitation was 591 mm. The individual months were characterized by the following parameters: January (t = −1.9 °C, pp = 29 mm), February (t = −0.9 °C, pp = 28 mm), March (t = 2.7 °C, pp = 33 mm), April (t = 8.9 °C, pp = 41 mm), May (t = 13.9 °C, pp = 69 mm), June (t = 17.4 °C, pp = 65 mm), July (t = 19, 4 °C, pp = 84 mm), August (t = 18.9 °C, pp = 71 mm), September (t = 13.7 °C, pp = 60 mm), October (t = 8.4 °C, pp = 45 mm), November (t = 3.7 °C, pp = 35 mm), December (t = −0.4 °C, pp = 32 mm). The experiment was located on soil developed on medium loamy sands, with a humus content of about 1.4% and a bulk density of 1.5 g·cm$^3$. The ground water table is at a depth of 1.5–2.5 m, depending of the season.

### 2.2. Structure of the Experiment

The whole field experiment area was managed in the organic system, as practiced since 1999. The experimental site is a production field of 6.3 ha divided into 5 approximately equal plots on which crop rotation is carried out on a 5-year cycle. Diseases were tested on a winter wheat crop, which was in a different field each year. Within the wheat field, an area of about 0.3 ha was designated in which more than a dozen winter wheat varieties were tested. Up to 2018 there were 12 varieties, in later years the number was increased to 16. All varieties were grown in plots 3 m wide and 5 m long. The plots were separated by paths, which were cultivated with a hand cultivator. A total of 64 plots were delineated; varieties were distributed in 4 replications in random blocks.

### 2.3. Monitoring of Leaf Diseases

Fungal disease monitoring of winter wheat was performed in June and July. The progress of disease development on the wheat flag leaf was monitored for three weeks. The final assessment was made at the maximum level of plant infestation, shortly before the physiological leaf drying phase. At the location under study, this phase occurred in late June to mid-July. At least five flag leaves were taken from the center of each plot at distances of about one meter from each other. Disease infestation was assessed visually on a on a percentage scale [%], while identifying individual diseases. Brown and yellow rust were evaluated separately from other diseases (Figure 1). Tan spot, powdery mildew, and

Fusarium were assessed based on the same scale as Septoria (Figure 2). Monitoring was carried out annually for 5 years in row, from 2018 to 2022.

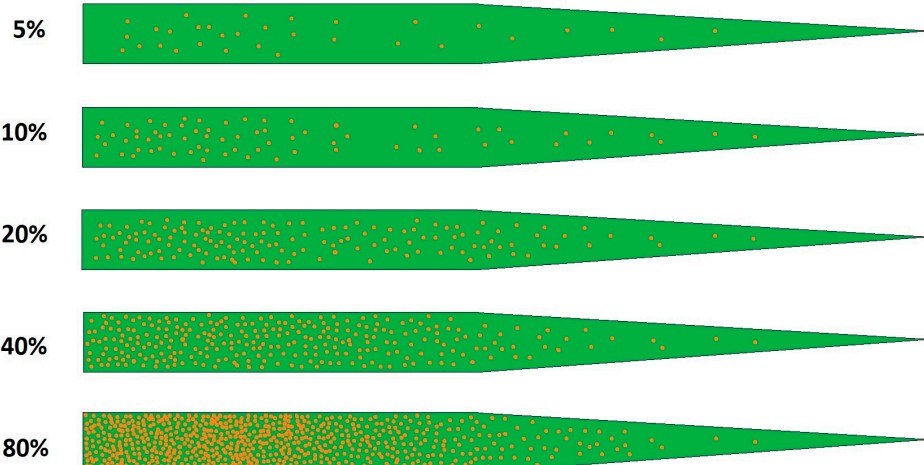

**Figure 1.** Simplified scale of brown rust infestation on wheat flag leaf, based on the percentage coverage of the leaf by uredia.

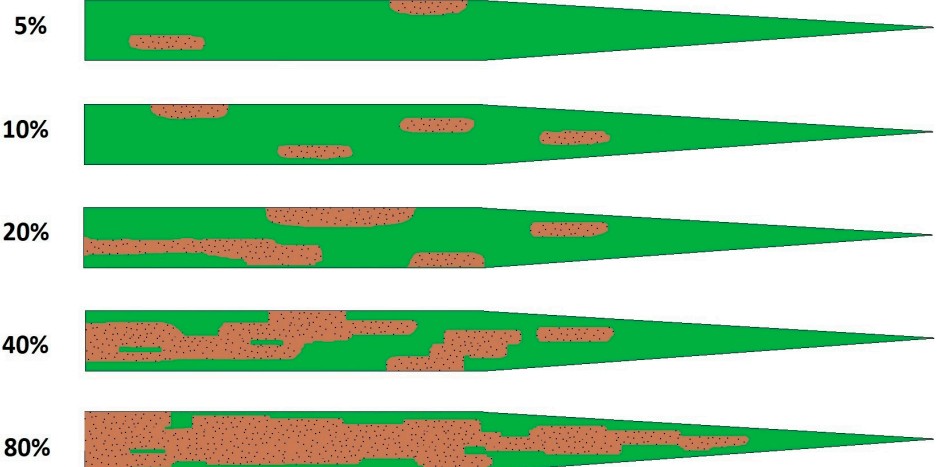

**Figure 2.** Simplified scale of Septoria infestation wheat flag leaf, based on the percentage coverage of the leaf by spots.

### 2.4. Weather Monitoring

Weather data were collected using a certified by National System of Weather Monitoring [8] weather station located at a direct distance of 200 m from the research plots. Total precipitation, mean temperature at daily level, as well as sunshine intensity and wind speed were obtained. Annual and monthly precipitation totals and annual and monthly mean temperatures were used for the analyses. Weather monitoring was carried out from January 2017 until the end of 2022. Based on the results, the evapotranspiration value ETP [mm] was calculated to estimate the monthly and annual climatic water balance CWB [mm]. This is a commonly calculated indicator of agricultural drought in Poland [24].

$$CWB = P - ETP$$

P—precipitation [mm]
ETP—evapotranspiration [mm]

The value of potential evapotranspiration is calculated using the simplified Penman formula:

$$ETP = 161 + 19.57d - 152.7\ln d + 0.0004034h^2 + 0.00186(t+5)^3 \\ + 0.004192(100-f)^2 + 0.0003681v(100-f)^{2.5}$$

(1)

d—length of day [h]
ln—natural logarithm
h—sunshine duration [h]
t—average air temperature [°C]
f—relative humidity, 1 p.m. [%]
v—average wind speed [m·s$^{-1}$].

*2.5. Statistical Processing*

The results of the fungal disease monitoring were presented by averaging across varieties in each year. The total infestation and the proportion of disease species identified were analyzed separately. Meteorological data and calculated indices were analyzed and presented at the level of sums and monthly averages. The results of individual varieties were compared using a multi-sample ANOVA with Tukey's post hoc test. The aim of the analysis was to identify the varieties with the lowest fungal disease infestation and to distinguish a homogeneous group of varieties with no statistical difference in the results obtained. The performance of the varieties was evaluated on the basis of the mean total infestation value obtained, which was assumed to be as low as possible, preferably twice as low as the mean value for all varieties. The representativeness of the samples was also taken into account. Results of varieties with too few replications or results with too large a standard error, exceeding 10% of the mean value, were rejected. The relationship between disease incidence and environmental factors was tested using Pearson's linear correlations. Only results with a high probability of $p < 0.001$ and correlations with $r > 0.20$ were considered. Statistical calculations were performed with the PAST version 4.11 software [25].

## 3. Results and Discussion

*3.1. Effect of the Air Temperature on Plant Health and Severity of Fungal Deses*

The weather pattern during the plant growth period was highly variable. Monitoring was carried out from 2017 to the end of 2022, and the period from August 2017 to July 2022 was included in the analyses. August was the warmest month of 2017 with an average of 19.7 °C, which may have affected water scarcity in following months. September, on the other hand, was relatively cool with an average temperature of 14.1 °C. The opposite was true in 2018, when the temperatures of both these months were high at 20.7 °C and 15.7 °C, which could have resulted in poor seed emergence. However, on the other hand, it could prevent voluntary cereal and weeds from emerging as well. In the years 2019 and 2020, average August temperatures were slightly lower than in 2019, while September temperatures were higher. In August and September of 2021, temperatures were exceptionally low (17.1 °C and 13.0 °C). At the tillering stage, average temperatures in Osiny reached 14.1 °C in September and 9.5 °C in October. Average October temperatures were the lowest in 2017 and 2021. The coldest November was reported in 2018. The wintering period for wheat lasted through December, January, and February in that region. The warmest winter was reported in 2017/2018 with an average temperature of 4.0 °C. The coldest winter was measured in 2021/2022 at −2.6 °C. Winters in the remaining years were characterized by slightly positive temperatures on average. March average temperatures were positive, with the lowest measured in 2018 (0.2 °C) and the highest in 2019 (5.5 °C). The warmest May was recorded in 2018 with 17.2 °C. Other years had May temperatures in the range of 11–14 °C. The hottest June was reported in 2019 (21.5 °C). In the other years,

the temperatures of June were similar, around 18.5 °C. The warmest July was recorded in 2021 (Table 1).

**Table 1.** Average monthly and annual temperatures in Osiny for the period 2017–2022.

| | Average Air Temperature [°C] | | | | | |
|---|---|---|---|---|---|---|
| | **2017** | **2018** | **2019** | **2020** | **2021** | **2022** |
| January | −4.6 | 0.4 | −2.6 | 1.8 | −1.6 | 0.5 |
| February | −1.1 | −3.5 | 2.7 | 3.4 | −2.7 | 2.9 |
| March | 6.0 | 0.2 | 5.5 | 4.5 | 2.9 | 2.7 |
| April | 7.5 | 13.6 | 9.6 | 8.3 | 6.5 | 6.5 |
| May | 13.9 | 17.2 | 13.0 | 11.0 | 12.6 | 13.4 |
| June | 18.1 | 18.8 | 21.5 | 18.3 | 18.6 | 18.7 |
| July | 18.6 | 20.7 | 18.7 | 18.6 | 21.9 | 19.1 |
| August | 19.6 | 20.7 | 20.2 | 20.0 | 17.0 | 20.8 |
| September | 14.1 | 15.7 | 14.5 | 15.1 | 13.1 | 11.8 |
| October | 9.5 | 10.3 | 10.9 | 10.6 | 9.2 | 10.7 |
| November | 4.6 | 3.9 | 6.4 | 5.4 | 5.1 | 3.9 |
| December | 2.5 | 1.0 | 3.1 | 1.7 | −1.1 | 0.2 |
| Annual avr. | 9.1 | 10.0 | 10.3 | 9.9 | 8.5 | 9.2 |

Numerous significant correlations were found between weather patterns and the severity of fungal diseases. Disease incidence could have been influenced by the mean annual air temperature, which varied greatly between 2017 and 2022. Disease incidence such as Septoria (r = 0.35) and total plant infection (r = 0.23) were correlated with high mean annual air temperature. The correlations can be considered weak but significant ($p < 0.001$). The correlations were different in the first wheat growing season and in the second, which were separated by winter. Plant health can also be affected by the previous year in the period between August and December. Conditions favorable to disease development on volunteer cereals and grasses may influence higher infestation of wheat at emergence in September and tillering phase in October. For overall plant infestation, there was a negative correlation with temperatures in August (r = −0.32), September (r = −0.33), October (r = −0.55), November (r = −0.35), and December (r = −0.29). The temperature of January and February did not significantly affect the overall disease infestation, while the temperature of March was negatively correlated with this factor (r = −0.39). The correlations presented are difficult to justify. This would imply that the warm conditions of the second half of the year in which the start of winter wheat vegetation occurs, as well as the warm spring, have a positive effect on the overall resistance of winter wheat. It would seem that these conditions should have a beneficial effect on plant diseases by ensuring that their intermediate hosts are growing. It is possible that the warm autumn and spring were linked to drought, which limited the vegetation of volunteer wheat and grasses after harvest. However, this statement needs further evidence. The occurrence of brown rust was mainly negatively correlated with the warm autumn and warm late summer: August r = −0.41, September r = −0.37, October r = −0.55, November r = −0.32, and December r = −0.39. Additionally, to some extent this was also true in case of Septoria: temperature in October r = −0.27 and in November r = −0.33. It is difficult to justify why the high temperatures of the autumn of the previous year would affect the lower infestation in the following year, especially as the monitoring took place in June and July. The occurrence of diseases other than Septoria and brown rust was strongly correlated with this factor, probably only because they were more likely to develop in years where there was little rust. The number of diseases found on leaves was positively correlated with the temperatures of

August r = 0.65, September r = 0.55, October r = 0.77, November r = 0.42, and December r = 0.68 of the year prior to monitoring. All correlations are statistically significant and too strong to be random. With the exception of wheat tan spot, these diseases were not relevant for overall plant health. It seems that a warm period at the time of sowing of winter wheat has a positive effect on its resistance. It should be noted that such correlations were not expected to be found in the previous year, as wheat vegetation only occurs in September, October, and November. A completely opposite relationship occurred in the following year. With the exception of a warm March, which works on the same principle as a warm autumn, the other months of the growing season had a negative effect on plant health, if the temperatures are higher. Overall infestation increased in years where high temperatures of April (r = 0.30) and May (r = 0.46) were recorded. Brown rust was most favored by a warm May (r = 0.38). Septoria was positively affected by the entire warm quarter, including April (r = 0.47), May (r = 0.43), and June (r = 0.20). Other diseases, on the other hand, were negatively correlated with warm conditions in May: tan spot r = −0.27, and fusariosis and other diseases r = −0.35, i.e., analogous to a warm autumn (Table 2). As in the previous case, the occurrence of tan spot, powdery mildew, and fusariosis may be due to their different thermal preferences or to avoiding the competition of more invasive diseases. Those diseases were also more prevalent if preceding year was warm and dry. This may be due to the fact those pathogens can be transmitted through crop residues that did not decompose properly [26]. Average July temperatures were no longer taken into account because disease monitoring was already performed at the very beginning of this month. An increase in temperature, especially in the second quarter of the year, can result in an increased risk of cereal disease infestation, regardless of the individual resistance of the varieties. High temperatures of the second half of the year seems to be favorable for winter wheat overall health.

**Table 2.** Pearson's linear correlation between average air temperature and wheat foliar disease infestation.

| Average Temperature [°C] | Septoria | Tan Spot | Brown Rust | Yellow Rust | Powdery Mildew | Fusariosis | All Diseases | Number of Diseases |
|---|---|---|---|---|---|---|---|---|
| August | 0.03 * | 0.25 | −0.41 | −0.16 | −0.06 | 0.64 | −0.32 | 0.65 |
| September | −0.03 * | 0.16 | −0.37 | −0.14 | −0.06 | 0.52 | −0.33 | 0.55 |
| October | −0.27 | 0.36 | −0.55 | −0.10 | 0.04 * | 0.75 | −0.55 | 0.77 |
| November | −0.33 | 0.29 | −0.32 | 0.03 * | 0.12 | 0.42 | −0.35 | 0.42 |
| December | 0.03 * | 0.31 | −0.39 | −0.13 | −0.02 * | 0.69 | −0.29 | 0.68 |
| January | −0.02 * | 0.12 | 0.05 * | 0.06 * | 0.07 | −0.03 * | 0.07 | −0.07 |
| February | −0.09 | 0.06 * | 0.07 | 0.06 * | 0.02 * | −0.27 | 0.03 * | −0.33 |
| March | −0.20 | 0.18 | −0.26 | −0.04 * | >0.01 * | 0.18 | −0.29 | 0.17 |
| April | 0.47 | −0.04 * | 0.15 | −0.10 | −0.13 | −0.02 * | 0.30 | −0.05 * |
| May | 0.43 | −0.27 | 0.38 | −0.01 * | −0.10 | −0.35 | 0.46 | −0.36 |
| June | 0.20 | −0.11 | 0.09 | −0.07 * | −0.11 | −0.19 | 0.12 | −0.21 |
| July | −0.08 | −0.04 * | −0.13 | −0.03 * | 0.02 * | 0.30 | −0.16 | 0.38 |
| Year | 0.35 | 0.03 * | 0.11 | −0.07 * | −0.11 | −0.12 | 0.23 | −0.19 |

* non-significant correlations with $p > 0.01$.

### 3.2. Influence of Precipitation on the Incidence of Fungal Diseases

In the surveyed region of Poland, winter wheat sowing takes place in early September. In 2017, the plants were very well supplied with water at the emergence stage, with a total amount of 212.8 mm. The opposite situation was in 2018, when rainfall in August and September was very low, accounting only for 75.7 mm. In 2019 and 2020, precipitation during this period was slightly higher (146.0 mm and 171.3 mm). In August and September of 2021, precipitation was exceptionally high, totaling 280.9 mm. Another very important

phase of winter wheat development is tillering, which in this region of Europe takes place in late October and early November. During this period, the highest rainfall occurred in 2017 (150.1 mm). In all subsequent years, with the exception of 2020, rainfall during this period was significantly lower. The wintering period for wheat extended into December, January, and February. Winter conditions varied greatly through the years of monitoring. For example, in the winters of 2017/2018 and 2020/2021, total precipitation amounted 50–100 mm, in the other winters it was 100–150 mm. Precipitation in March was similar in 2018–2020, while it was twice as low in 2021 and 2022 (Table 3). The reverse was the case in April, with rainfall averaging higher in 2021 and 2022. May of 2021 recorded exceptionally high rainfall, while exceptionally low precipitation was recorded in May of 2022. June is one of the most important months for winter wheat development, as stalk shooting and the fastest growth occur during this period. Only in 2020 was high rainfall recorded in this month; in 2021 it was average, while in the other years it was low for this region. This factor may have influenced the exceptionally high plant resistance in 2020 and 2021. The precipitation in July was no longer so important for disease incidence, as the assessment was made in late June or early July. Shortly thereafter, the last leaves of wheat naturally dried out. Interestingly, in almost all years, precipitation in July reached a sum of 100 mm. The only exceptions were the years of 2019 and 2020; however, this did not have an impact on diseases. Considering the scale of a whole year, the most rain fell in 2020 (780.8 mm), 2017 (766.5 mm), and in 2021 (693.4 mm). In 2018, the annual sum of perception was only 501.5 mm, in 2019 it was only 540.4 mm, and it was the same in 2022 (540.1 mm). This is well below the annual average for the region, with is 591 mm [8]. There was also no year where an exact average rainfall total value occurred. Therefore, the results can be divided into two groups, related to high and low annual precipitation.

**Table 3.** Monthly and annual precipitation sums in Osiny for the period of 2019–2022.

| | Total Precipitation [mm] | | | | | |
| --- | --- | --- | --- | --- | --- | --- |
| | **2017** | **2018** | **2019** | **2020** | **2021** | **2022** |
| January | 4.5 | 17.3 | 34.9 | 35.3 | 35.2 | 48.3 |
| February | 53.3 | 17.1 | 17.1 | 61.8 | 24.9 | 41.1 |
| March | 32.9 | 31.2 | 22.7 | 26.3 | 12.8 | 12.1 |
| April | 71.7 | 29.8 | 35.4 | 11.9 | 51.7 | 43.6 |
| May | 67.5 | 59.4 | 86.4 | 115.8 | 47.7 | 39.1 |
| June | 33.6 | 38.1 | 38.6 | 183.2 | 61.6 | 26.6 |
| July | 119.6 | 122.5 | 33.6 | 50.6 | 108.6 | 92.4 |
| August | 107.6 | 27.7 | 86.7 | 51.9 | 218.5 | 42.1 |
| September | 105.2 | 48 | 59.3 | 119.4 | 62.4 | 75.0 |
| October | 94.9 | 40.5 | 40.9 | 88.5 | 5.2 | 28.8 |
| November | 55.2 | 8.9 | 30 | 12.1 | 38.6 | 36.8 |
| December | 20.5 | 61 | 54.8 | 24 | 26.2 | 54.7 |
| Annual sum | 766.5 | 501.5 | 540.4 | 780.8 | 693.4 | 540.1 |

High and optimal mean annual sum of precipitation most likely had a positive effect on plant health. Annual rainfall total and overall plant infestation were found to be negatively correlated (r = −0.48). For individual diseases, the presence of brown rust (r = −0.43) and Septoria (r = −42) were negatively correlated with precipitation sum, while the presence of tan spot (r = 0.32), and the presence of fusariosis (r = 0.52) were positively correlated (Table 4). The annual sum of rainfall had no effect on the incidence of yellow rust. In particular, rainfall was tracked in the period prior to winter wheat vegetation. Precipitation in August, before sowing, was positively correlated with the occurrence

of brown rust (r = 0.41) in the following year. Conversely, precipitation in August was negatively correlated with the cover of tan spot (r = −20) and fusarium (r = −0.61) on flag leaf. A possible connection between rainfall in August and its alleged effect after ten months may be hard to justify. In this region, winter wheat is harvested at the beginning of or in mid-August, when most of the leaf diseases have long since disappeared. Instead, they have the chance to develop on volunteer cereal or on intermediate hosts. However, under Central European conditions, the importance of intermediate hosts is marginal and most diseases are transmitted directly from crop residues [27,28]. The role of high rainfall in August for brown rust remains unclear, especially as rainfall in subsequent months was already negatively correlated with its future occurrence. There was also no effect of rainfall in August on Septoria in the following year. Precipitation in October was negatively correlated with the occurrence of brown rust the following year (r = −0.24), but precipitation in November was positively correlated (r = 0.35). Winter wheat rust infection still occurs in autumn; however, it is difficult to assess whether high rainfall in November significantly contributes to this. Previously, cool autumn conditions were found to be indirectly associated to a higher incidence of rust, despite the fact that it is a rather thermophilic group of plant diseases. It is likely that the rain itself is more important in the spread of rust, and that the decrease of temperature is simply linked to any longer period of rainfall. The incidence of Septoria in June was also positively associated with November rainfall (r = −0.28). The results indicate that a cool and wet November may be the cause of disease incidence the following year. Precipitation totals in December were positively correlated with the subsequent occurrence of tan spot (r = 0.20), but negatively with brown rust (r = −20). Rainfall in January did not significantly affect the overall summer infestation; however, there was a negative correlation with the occurrence of Septoria r = −0.32, and other fungal diseases, mainly fusariosis. February precipitation was also negatively correlated with the incidence of Septoria (r = −0.25), but positively correlated with the incidence of tan spot (r = 0.23). Precipitation in March was, on the other hand, positively correlated with Septoria incidence (r = 0.34). This was probably due the fact that 2018 recorded the highest rainfall in March, while the rest of the period was dry. Precipitation in April was negatively correlated with the incidence of tan spot (r = −0.24) and other less important diseases (r = −0.21). Precipitation in May and June influenced lower overall plant infestation (r = −0.22, r = −0.36). The occurrence of brown rust was negatively correlated with rainfall in May (r = −0.30) and June (r = −0.39), while the occurrence of Septoria was negatively affected by high rainfall in June (r = −0.20). The occurrence of tan spot was positively related to rainfall in May (r = 0.31) and June (r = 0.37), as was the occurrence of other rarer diseases, which were difficult to identify (r = 0.41, r = 0.54).

With the exception of May and June, the correlation results appear weak and random. Spring water supply can affect the fitness and resistance of plants in the early stages of growth. It may also help the spread of diseases at certain stages of the pathogen's life cycle. Finally, it can indirectly create conditions that favor the development of plant diseases. With few exceptions, higher rainfall sums in the first half of the year have been found to have a positive effect on plant health, increasing resistance to Septoria and brown rust. However, they promote other diseases such as tan spot and fusariosis. This is still a beneficial situation, as tan spot and other diseases are not as important as Septoria and brown rust in this location. Other authors found different patterns in various European regions, including Western Poland. Major fungal diseases affected the flag leaf of winter wheat more severely in the rainy year of 2017 than in the hot and dry year of 2018 [29]. However, the research was not performed under an organic system, and crop protection was used. Our results indicate that providing adequate water supply to organic wheat crops in early summer can ensure not only good yield, but also greater resistance to major fungal diseases.

**Table 4.** Pearson's linear correlation between sums of precipitation and wheat foliar disease infestation.

| Sum of Precipitation [mm] | Septoria | Tan Spot | Brown Rust | Yellow Rust | Powdery Mildew | Fusariosis | All Diseases | Number of Diseases |
|---|---|---|---|---|---|---|---|---|
| August | 0.01 * | −0.20 | 0.41 | 0.15 | 0.06 * | −0.61 | 0.35 | −0.64 |
| September | −0.06 | −0.01 * | −0.14 | −0.03 * | 0.02 * | 0.34 | −0.14 | 0.41 |
| October | 0.08 | 0.08 | −0.24 | −0.11 | −0.04 * | 0.53 | −0.18 | 0.57 |
| November | 0.28 | −0.12 | 0.35 | 0.06 * | −0.01 * | −0.36 | 0.41 | −0.40 |
| December | −0.01 * | 0.20 | −0.20 | −0.07 | −0.04 * | 0.19 | −0.16 | 0.14 |
| January | −0.32 | −0.07 | 0.09 | 0.13 | 0.10 | −0.35 | −0.05 * | −0.34 |
| February | −0.25 | 0.23 | −0.15 | 0.06 * | 0.10 | 0.12 | −0.18 | 0.08 |
| March | 0.34 | 0.13 | −0.04 * | −0.11 | −0.10 | 0.20 | 0.12 | 0.15 |
| April | −0.12 | −0.24 | 0.12 | 0.05 * | 0.02 * | −0.21 | 0.02 * | −0.15 |
| May | >0.01 * | 0.31 | −0.30 | −0.09 | −0.02 * | 0.41 | −0.22 | 0.36 |
| June | −0.20 | 0.37 | −0.39 | −0.04 * | 0.06 | 0.54 | −0.36 | 0.52 |
| July | 0.04 * | −0.16 | 0.15 | 0.05 | 0.03 * | −0.07 | 0.13 | −0.03 * |
| Year | −0.42 | 0.32 | −0.43 | 0.01 | 0.12 | 0.52 | −0.48 | 0.54 |

* non-significant correlations with $p > 0.01$.

*3.3. Climatic Water Balance and Plant Health*

The climatic water balance CWB [mm] has been forged as an indicator of agricultural drought. For winter cereals on this soil category, the threshold value for drought is a water balance between −150 mm and −241 mm, by over a 60-day monitoring period (Table 5) [8].

**Table 5.** Drought threshold in a two-month period for winter wheat on soils in the second drought susceptibility category.

| Period | Threshold [mm] |
|---|---|
| 21.III–20.V | −150 |
| 1.IV–31.V | −153 |
| 11.IV–10.VI | −158 |
| 21.IV–20.VI | −166 |
| 1.V–30.VI | −175 |
| 11.V–10.VII | −186 |
| 21.V–20.VII | −201 |
| 1.VI–31.VII | −220 |
| 11.VI–10.VIII | −241 |

Simply put, this means that if, for example, the total climatic water balance for April and May is less than −153 mm, this indicates a legitimate agricultural drought [8]. Therefore, we can conclude that in none of the years studied was there a true drought in the April–May period. The closest to a real drought was April and May of 2018 with a CWB of −148 mm (Table 6). In the following months, precipitation amounts are higher, so in order to consider this period as a true drought, the CWB value must be even lower. In the period from May to the end of June, the climatic water balance must be below −175 mm, and below −220 mm in the June–July period. Considering this assumption, in the May–June period, drought occurred in 2022 with a CWB of −205 mm, while on the borderline of values were found in May and June of 2018: CWB = −173 mm (Table 4). The years of 2018 and 2022 can be considered the only in which severe agricultural drought occurred for

winter wheat. These years also saw the lowest plant resistance, with an average overall infestation of 42.3% in 2018 and 36.9% in 2022. In spite of the fact that real drought only occurred a few times during the study, water supply was insufficient for optimal plant growth in most of the monitoring seasons.

**Table 6.** Monthly and annual climatic water balance in Osiny for the period 2017–2022.

| | Climatic Water Balance [mm] | | | | | |
|---|---|---|---|---|---|---|
| | **2017** | **2018** | **2019** | **2020** | **2021** | **2022** |
| January | 10.6 | 28.7 | 47.4 | 44.5 | 47.1 | 59.1 |
| February | 48.1 | 11.7 | 7.3 | 53.9 | 17.6 | 32.5 |
| March | 4.4 | 3.2 | −11.4 | −14.8 | −18.0 | −33.2 |
| April | 11.5 | −64.7 | −45.8 | −82.2 | −12.6 | −12.5 |
| May | −42.7 | −83.1 | −2.4 | 18.3 | −54.7 | −82.1 |
| June | −106.8 | −90.6 | −130.5 | 62.9 | −89.2 | −123.4 |
| July | −4.4 | −4.1 | −93.5 | −79.5 | −33.3 | −27.9 |
| August | −12.9 | −99.0 | −27.6 | −66.3 | 128.3 | −82.3 |
| September | 54.0 | −21.3 | −4.7 | 49.9 | 3.7 | 23.8 |
| October | 74.7 | 1.7 | 3.4 | 65.2 | −32.1 | −1.9 |
| November | 60.5 | 7.9 | 32.8 | 15.8 | 39.9 | 42.6 |
| December | 37.1 | 78.2 | 70.4 | 39.9 | 78.6 | 71.3 |
| Annual | 134.1 | −231.4 | −154.6 | 107.6 | 75.2 | −134.0 |

Overall plant infestation was negatively correlated with water balance throughout the first half of the year: January r = −0.35, February r = −0.32, March r = −0.31, April r = −0.29, May r = −0.17, June r = −0.15, and July r = −0.10, with a *p*-value < 0.001 in all above correlations (Table 7). As we can see, poor water balance has a negative impact on plant health from the beginning of the year. Interestingly, the overall plant infestation was positively correlated with the annual precipitation (r = 0.47) and also with the good water balance in the following months of the previous year: October r = 0.22, November r = 0.28, and December r = 0.42. This confirms the results from weather monitoring data, where combination of precipitation amounts and average day temperatures affected plant health. The results obtained indicate that the dry and hot first half of the year and the wet and cool second half create favorable conditions for the development of some winter wheat diseases. This could also explain the greater resistance of spring wheat, as it is not exposed to unfavorable conditions in autumn. At the same location, a significantly lower overall infestation for spring wheat of 21.7% was obtained [23]. The situation was different for individual diseases. The occurrence of Septoria was positively correlated with a positive CWB in November, r = 0.28. The occurrence of brown rust was positively correlated with a positive CWB in all months of the preceding year, especially August r = 0.50, November r= 0.35, and December r = 0.26. The water balance in the months of the harvest year was negatively correlated with the severity of brown rust, May r = −0.39 and June r = −0.37, as it was for the water balance of the whole year r = −0.41. This means that the drought occurring in May and June significantly favors the gradation of rust, while the other months can affect its spread if they are rainy. The incidence of tan spot was positively affected by a positive annual water balance of r = 0.35, especially in case of May (r = 0.33) and June (r = 0.35). In contrast, a positive water balance in April was negatively correlated with tan spot cover on leaf in the end of June (r = −0.28). This would suggest that an early drought could favor the development of tan spot; however, the following months should be wet. No significant correlations were found for the occurrence of powdery mildew and yellow rust. For the other fungal diseases, comprising mainly cases of leaf fusariosis, a correlation was

found between positive water balance in May (r = 0.52) and June (r = 0.52), and throughout the whole year (r = 0.55). September and October of the previous year were also positively correlated with fusariosis occurrence (r = 0.27, r = 0.57). A positive CWB in some months was negatively correlated with fusariosis and other pathogens: August r = −0.76, January r = −0.54, December r = −0.48, November r = −0.43. It is difficult to assess whether the previous year's drought can influence the infestation of these diseases, it may rather prevent the occurrence of brown rust and Septoria, which are strongly competitive and cover the most of the leaf area.

**Table 7.** Pearson's linear correlation between climatic water balance (CWB [mm]) and wheat foliar disease infestation.

| CWB [mm] | Septoria | Tan Spot | Brown Rust | Yellow Rust | Powdery Mildew | Fusariosis | All Diseases | Number of Diseases |
|---|---|---|---|---|---|---|---|---|
| August | 0.11 | −0.19 | 0.50 | 0.15 | 0.03 * | −0.76 | 0.07 | −0.72 |
| September | −0.11 | −0.09 | −0.12 | −0.03 * | 0.02 * | 0.27 | 0.16 | 0.33 |
| October | −0.09 | 0.03 * | −0.32 | −0.10 | −0.01 * | 0.57 | 0.22 | 0.58 |
| November | 0.28 | −0.05 * | 0.35 | 0.06 | −0.01 * | −0.43 | 0.28 | −0.44 |
| December | 0.11 | 0.01 * | 0.26 | 0.07 | −0.01 * | −0.48 | 0.42 | −0.51 |
| January | −0.17 | 0.17 | 0.27 | 0.13 | 0.06 * | −0.54 | −0.35 | −0.47 |
| February | −0.09 | 0.25 | −0.15 | 0.02 * | 0.06 * | 0.15 | −0.32 | 0.06 |
| March | 0.19 | 0.16 | −0.27 | −0.14 | −0.07 | 0.52 | −0.31 | 0.45 |
| April | −0.13 | −0.28 | 0.19 | 0.07 | 0.02 * | −0.30 | −0.29 | −0.19 |
| May | −0.07 | 0.33 | −0.39 | −0.08 | 0.00 * | 0.51 | −0.17 | 0.40 |
| June | −0.10 | 0.35 | −0.37 | −0.05 * | 0.04 * | 0.52 | −0.15 | 0.40 |
| July | 0.04 * | −0.27 | 0.28 | 0.07 | 0.01 * | −0.32 | −0.10 | −0.22 |
| Year | −0.17 | 0.35 | −0.41 | −0.04 * | 0.06 * | 0.55 | 0.47 | 0.44 |

\* non-significant correlations with *p* > 0.01.

### 3.4. Resistance of Winter Wheat in Different Years

The overall result of winter wheat, as well as individual varieties, differ from year to year. In 2018, the average flag leaf infestation was at the level of 42.3%. The varieties Estivus, Rokosz, and Belissa were the healthiest; however, none of them achieved satisfactory resistance (Table 8). Therefore, a completely different set of varieties was tested in 2019. The choice proved to be the right one, as a lower leaf infestation of 27.3% was achieved, under weather conditions comparable to 2018. The varieties Plejada and RGT Kilimanjaro obtained the best results. The weakest variety in 2019 was Memory, which was swapped in the following year for the variety Artist. This was a right choice as it proved to be one of the more resistant and best yielding varieties in the following years. In the next two vegetation seasons, the choice of varieties and weather conditions resulted in very good winter wheat performance. A total of 12 varieties in 2020 and a total of 15 in 2021 achieved resistance in excess of 85% (up to 15% infestation) at the very late evaluation date. The vegetation of the plants was extended, and the flag leaves remained practically healthy until the physiological desiccation stage. Very satisfactory resistance was achieved during this period, but the varieties tested had a rather average yield. Therefore, another set of varieties, known for their good grain yield, was tested in 2022. However, with the exception of the varieties Symetria and Comandor, acceptable resistance was not achieved. In addition, conditions in 2022 resembled those in 2018–2019, which did not allow the new set of varieties to be tested under optimal weather. Particularly poor results were obtained for 2018 and 2022. There are two reasons for the situation. Firstly, both 2018 and 2022 were characterized by very low annual precipitation. Secondly, the selection of varieties may

not have been as accurate when compared to 2019–2021. Both years were also preceded by wet autumns in 2017 and in 2021, which could have negative effect, based on previous analyses. Although the year 2019 was very different from 2020–2021, similar results were obtained. The year 2019 was characterized by low precipitation and high temperatures, while 2020 and 2021 recorded the highest precipitation in the last decade and relatively moderate temperatures. As a result, the plants were very healthy and resilient, especially to Septoria and brown rust.

**Table 8.** Average fungal disease infestation on a flag leaf of different varieties of winter wheat in different years.

| 2018 | 2019 | 2020 | 2021 | 2022 |
|---|---|---|---|---|
| Estivus–20.9% | Plejada *–13.5% | Artist *–9.6% | Plejada *–6.8% | Symetria–10.0% |
| Rokosz–22.4% | RGT Kilimanjaro *–18.4% | Plejada *–9.8% | Hybery *–7.2% | Comandor *–12.2% |
| Belissa–22.8% | Formacja–21.3% | Delawar *–10.2% | RGT Kilimanjaro *–8.6% | KWS Universum–23.4% |
| Hondia–32.6% | Hybery *–21.5% | Hybery *–10.3% | Euforia–9.2% | Argument–25.4% |
| Pokusa–36.9% | Comandor–* 21.9% | RGT Kilimanjaro *–10.6% | RGT Bilanz *–9.3% | Impresja–28.0% |
| Markiza–44.1% | Hondia–22.8% | Owacja *–11.1% | Artist *–9.4% | Ambicja–28.1% |
| KWSOzon–44.4% | Delawar *–23.0% | KWS Spencer *–11.9% | Hondia–9.6% | SY Yukon–29.9% |
| Ostka Strzelecka–49.4% | Euforia–23.5% | RGT Bilanz *–12.3% | Comandor *–9.7% | MHR Promienna–34.0% |
| Jantarka–52.2% | Medalistka–23.5% | Euforia–12.7% | Bonanza *–10.0% | RGT Provision–34.3% |
| Fidelius–57.3% | Bonanza *–24.9% | Comandor *–13.4% | Owacja *–10.5% | Euforia–36.1% |
| Linus–61.4% | KWS Spencer *–30.5% | Bonanza *–14.3% | Delawar *–10.6% | SY Dubaj–38.4% |
| Arktis–63.3% | RGT Bilanz–33.6% | Medalistka–14.9% | Formacja–11.3% | Formacja–45.5% |
| - | Ostka GG–35.0% | Formacja–15.7% | KWS Spencer *–11.6% | SY Orofino–47.1% |
| - | Owacja *–35.3% | Hondia–17.3% | Medalistka–12.1% | Kariatyda–60.0% |
| - | Tytanika–38.6% | Ostka GG–20.0% | Tytanika–13.5% | LG Keramik–67.4% |
| - | Memory–49.9% | Tytanika–23.5% | Ostka GG–15.9% | Almari–70.9% |

* best performing varieties in long-term analysis.

### 3.5. Ranking of Winter Wheat Varieties Based on Fungal Disease Resistance

A total of 41 varieties of winter wheat were tested between 2018 and 2022. Depending on the variety, between 16 and more than 100 replications were obtained. A change of the variety set was made in 2019, also similarly in 2022. In effect, some varieties were tested only in 2018 (Arktis, Belissa, Estivus, Fidelius, Jantarka, KWS Ozon, Linus, Markiza, Ostka Strzelecka, Pokusa, and Rokosz), others only in 2022 (Almari, Ambicja, Argument, Impresja, Kariatyda, KWS Universum, LG Keramik, MHR Promienna, RGT Provision, SY Dubai, SY Orofino, SY Yukon, and Symetria). There were also varieties that were tested for four consecutive years: Comandor, Euphoria, Formation, and Hondia (Table 9). The remaining varieties were grown for at least two or three years. For the winter wheat studied, the average grain yield was 4.65 t·ha$^{-1}$, while the average weight of one thousand grains (TGW) was 39.88 g. As it turned out, in the multi-year perspective, the resistance of winter wheat was not directly related to the yield. The average yield of the 10 healthiest varieties was 4.81 t·ha$^{-1}$ and average weight of one thousand grains was 39.30 g, while the best-yielding varieties had a yield of 5.98 t·ha$^{-1}$ and grain mass of 43.99 g. The resistance of these varieties was at the average level for the studied population of winter wheat. The highest yielding varieties had an average fungal disease infestation of 28.3%. This was

almost twice as high as the ten healthiest varieties. The highest yielding varieties included: SY Yukon, Ambition, Symetria, SY Orofino, RTG Provision, Artist, Kariatyda, Argument, KWS Universum, and Medialistka. The varieties differed in their overall resistance to fungal diseases. The average infestation of varieties was at the level of 30.3% (69.7% of healthy assimilation area on flag leaf) and 25 varieties scored below this value, while 12 varieties had a promising score of less than 15% of infestation. The least infested varieties had an average value below 10%. This means that almost the entire flag leaf was healthy up to the physiological desiccation stage. The healthiest 11 varieties were not statistically different in their results, achieving an average between 8.6% and 13.5%. However, some of the varieties, such as Symetria, also had a wide confidence range (SE) constituting 30% of the mean value. This resulted from the wide spread of the obtained data or from a small number of replications $n = 20$ (Table 9). For the low-ranking varieties, statistical differences between results were more frequent, but not as often as the mean value would indicate. The weaker varieties had a large scatter of results, leading to a high standard error value, which exceeded 20% of the mean value. This situation also occurred when not enough replicates were obtained, for example, when a variety was only tested in one year. Consequently, there were fewer statistically significant differences between the tested varieties than expected. The difference between individual varieties was not essential for this study. The aim of the work was not to compare whole sets of varieties with one another, but to select the best performing group without statistical differences between the chosen varieties. During the analyses, it was found that the number of repetitions and the number of years in cultivation did not affect the value of the infestation results; however, the varieties grown in 2019–2021 had the best results and better data distributions, with narrow confidence intervals. This was true for both more resistant and more susceptible varieties. Therefore, only the varieties that had low infestation and reliable results were included for the final evaluation. Reliability was assessed by a narrow confidence range of SE < 10% of average value for the variety tested (Table 9).

**Table 9.** List of tested varieties ranked by lowest average fungal disease infestation.

| Variety | Number of Samples | Number of Years | Yield [t·ha$^{-1}$] | TGW * [g] | Average Infestation [%] | SE | SE to Avr. Ratio [%] |
|---|---|---|---|---|---|---|---|
| Plejada | 84 | 3 | 4.6 | 41.8 | 8.6 | 0.4 | 4.7 |
| Artist | 76 | 2 | 5.9 | 42.5 | 9.5 | 0.3 | 3.2 |
| Hybery | 83 | 3 | 5.4 | 38.3 | 9.8 | 0.7 | 7.1 |
| Symetria | 20 | 1 | 6.2 | 41.1 | 10.0 | 3.2 | 31.7 |
| RGT Kilimanjaro | 87 | 3 | 4.5 | 40.5 | 10.4 | 0.6 | 5.3 |
| Delawar | 81 | 3 | 4.9 | 34.6 | 11.7 | 0.7 | 6.1 |
| Comandor | 103 | 4 | 4.9 | 37.3 | 12.3 | 0.7 | 5.7 |
| RGT Bilanz | 85 | 3 | 4.2 | 38.3 | 12.9 | 1.0 | 7.5 |
| Owacja | 88 | 3 | 5.3 | 37.6 | 13.1 | 0.9 | 6.9 |
| Bonanza | 83 | 3 | 4.7 | 37.1 | 13.2 | 0.7 | 5.6 |
| KWS Spencer | 84 | 3 | 4.0 | 39.7 | 13.5 | 0.8 | 5.9 |
| Medalistka | 80 | 3 | 4.9 | 39.8 | 14.4 | 0.7 | 4.8 |
| Hondia | 91 | 4 | 4.2 | 42.1 | 15.7 | 0.9 | 5.9 |
| Euforia | 104 | 4 | 4.6 | 41.4 | 16.7 | 1.4 | 8.1 |
| Ostka GG | 86 | 3 | 3.6 | 41.3 | 19.5 | 1.3 | 6.8 |
| Tytanika | 84 | 3 | 4.3 | 32.5 | 20.0 | 1.3 | 6.7 |

**Table 9.** *Cont.*

| Variety | Number of Samples | Number of Years | Yield [t·ha$^{-1}$] | TGW * [g] | Average Infestation [%] | SE | SE to Avr. Ratio [%] |
|---|---|---|---|---|---|---|---|
| Formacja | 100 | 4 | 5.2 | 38.0 | 20.3 | 1.8 | 8.7 |
| Estivus | 16 | 1 | 4.1 | 38.6 | 20.9 | 3.2 | 15.2 |
| Rokosz | 16 | 1 | 2.6 | 39.9 | 22.4 | 3.3 | 14.7 |
| Belissa | 16 | 1 | 3.5 | 35.2 | 22.8 | 3.4 | 15.1 |
| KWS Universum | 20 | 1 | 5.6 | 43.9 | 23.4 | 3.9 | 16.6 |
| Argument | 20 | 1 | 5.7 | 44.9 | 25.4 | 3.5 | 13.8 |
| Impresja | 20 | 1 | 4.3 | 43.6 | 28.0 | 2.1 | 7.5 |
| Ambicja | 20 | 1 | 6.4 | 50.3 | 28.1 | 3.2 | 11.6 |
| SY Yukon | 20 | 1 | 6.6 | 44.2 | 29.9 | 2.2 | 7.3 |
| MHR Promienna | 20 | 1 | 4.7 | 41.2 | 34.0 | 4.5 | 13.1 |
| RGT Provision | 20 | 1 | 6.0 | 42.8 | 34.3 | 6.8 | 19.8 |
| Pokusa | 16 | 1 | 3.5 | 35.7 | 36.9 | 5.0 | 13.5 |
| SY Dubaj | 20 | 1 | 5.4 | 45.1 | 38.4 | 5.4 | 14.1 |
| Markiza | 16 | 1 | 3.2 | 35.1 | 44.1 | 4.7 | 10.7 |
| KWS Ozon | 16 | 1 | 3.3 | 38.7 | 44.4 | 5.2 | 11.6 |
| SY Orofino | 20 | 1 | 6.0 | 45.6 | 47.1 | 5.0 | 10.5 |
| Ostka Strzelecka | 18 | 1 | 3.6 | 35.9 | 49.4 | 4.0 | 8.1 |
| Memory | 16 | 1 | 4.2 | 35.5 | 49.9 | 5.4 | 10.9 |
| Jantarka | 18 | 1 | 4.2 | 42.7 | 52.2 | 7.3 | 1.4 |
| Fidelius | 16 | 1 | 4.2 | 38.2 | 57.3 | 5.1 | 8.8 |
| Kariatyda | 20 | 1 | 5.8 | 44.0 | 60.0 | 5.8 | 9.7 |
| Linus | 18 | 1 | 3.9 | 34.7 | 61.4 | 5.7 | 9.2 |
| Arktis | 16 | 1 | 3.1 | 32.6 | 63.3 | 4.6 | 7.2 |
| LG Keramik | 20 | 1 | 5.3 | 41.8 | 67.4 | 3.8 | 5.6 |
| Almari | 20 | 1 | 5.1 | 46.6 | 70.9 | 5.0 | 7.1 |
| Average | 45.3 | 1.9 | 4.7 | 40.0 | 30.3 | 3.1 | 9.9 |

\* TGW—thousand grain weight.

Finally, the ten most resistant varieties were selected, based on several criteria. Firstly, it was assumed that the overall fungal disease infection score should be at least twice as low as the multi-year average of the whole population (30.3%). Secondly, it was assumed that the score of the selected varieties should not be too different. Varieties whose score was significantly weaker than the best performing variety were rejected. Thirdly, it was ensured that the results obtained were reliable. Therefore, varieties that were only tested in one year, obtained too few replications, or had a large standard error exceeding 10% of the mean value, were rejected. Nevertheless, these varieties should be considered worthy of attention and further studied. As a result, more than half of the tested varieties met these objectives, so only the varieties with an average fungal disease infestation of less than 15% were selected for the final ranking. The top 10 varieties were Plejada 8.6%, Artist 9.5%,

Hybery 9.8%, Symetria 10%, RGT Kilimanjaro 10.4%, Delawar 11.7%, Comandor 12.3%, RGT Bilanz 12.9%, Owacja 13.1%, Bonanza 13.2%, and KWS Spencer 13.5% (Figure 3).

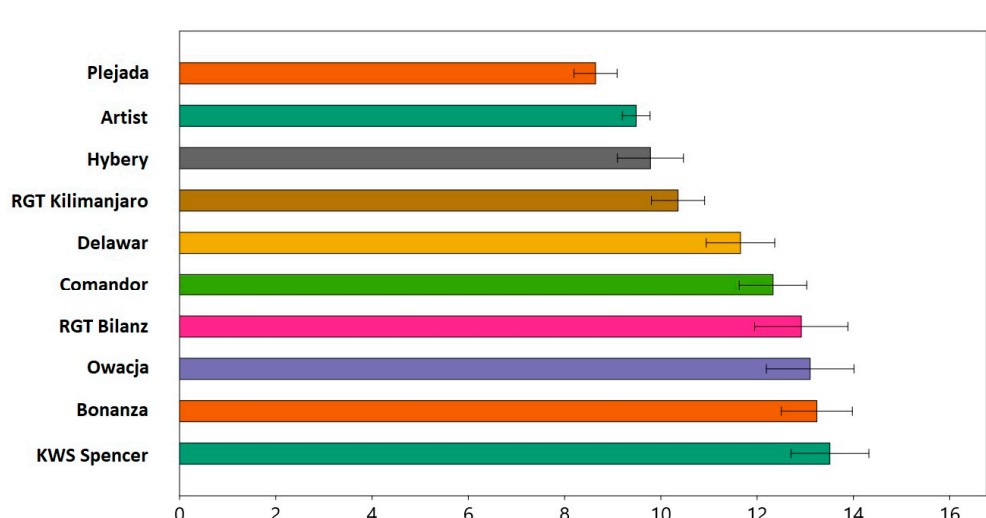

**Figure 3.** Selected 10 varieties most resistant to fungal diseases. Results are not different based on the ANOVA test of significance level $p < 0.01$.

Results shows the most resilient varieties in the five-year monitoring period in organic farming. The aim of this work was achieved, yet the results can only have a limited use. The aim itself may be contradictory to global needs. Numerous studies show the importance of wheat as the one of the main sources of human food. In order to feed a growing world population, it is estimated that grain production should be increased at least by 40% [30]. This will not be possible by using organic farming; that is not the most productive system for winter wheat. Additionally, methods that require manual monitoring on the field cannot provide a wide range of prevention systems for a considerable geographic area. For this purpose, molecular identification and prediction models need to be used [30,31]. In defense of basic methods, it must be stated that under the changing climate, new pathogens and nonspecific symptoms may occur, that can be detected only in the field by expert. Furthermore, even the most advanced modeling tools need much data to be fed and calibrated. A very promising method is remote sensing of fungal diseases infestation, alongside other biotic and abiotic stresses assessments [32].

### 3.6. Importance of Individual Fungal Diseases

It was observed that the most resistant varieties were characterized not only by low infestation but also by a high diversity of diseases on flag leaves. On average, the most resistant varieties had five or more fungal diseases at the same time. However, this should not be interpreted as a correlation between disease diversity and plant health. Generally low leaf infestations and unoccupied assimilation area allowed for the presence of other, less frequent diseases. Additionally, with considerable infestation by one pathogen, other symptoms were difficult to detect. Conversely, only one or two diseases were present on the least resistant varieties, mainly brown rust and Septoria spots (Figure 4).

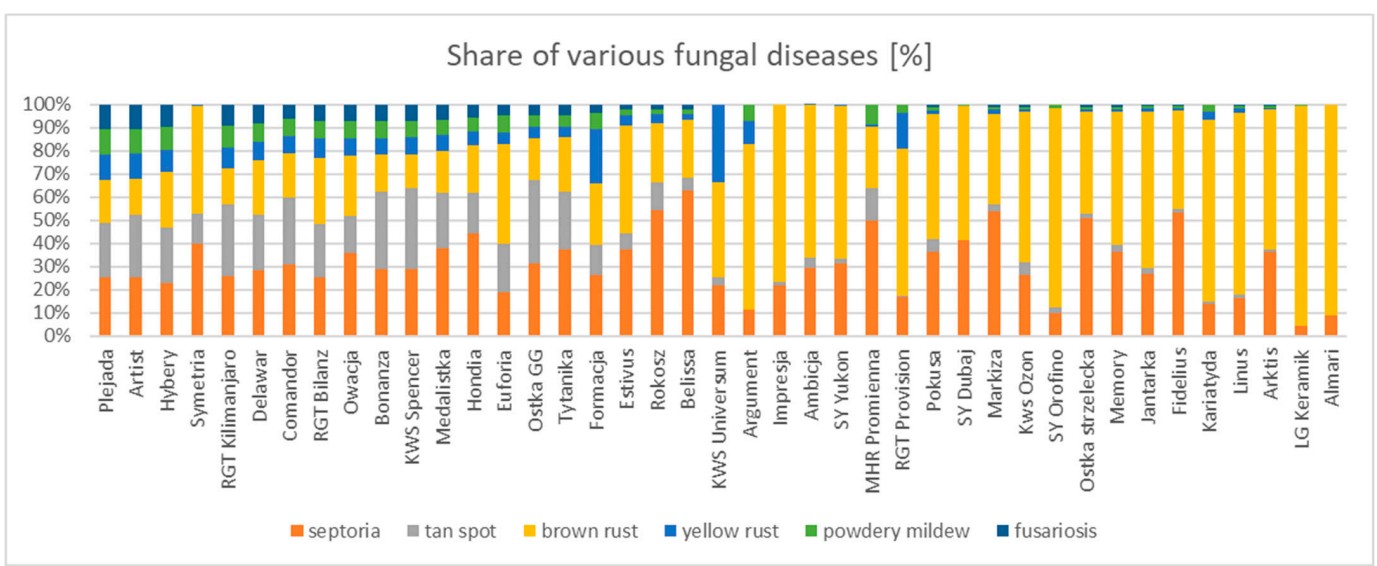

**Figure 4.** Share of individual diseases in the pool of total infestation on the tested varieties. The most resistant varieties have been placed on the left side of the graph, the most susceptible on the right.

The overall infestation was mainly driven by brown rust (r = 0.87, *p* < 0.001). The average leaf cover by this pathogen was about 17%, thus it was responsible for more than half of the total fungal disease infestation. It sometimes occurred as the only disease found, but it was usually accompanied by Septoria and tan spot. With a high severity of rust, other, rarer fungal diseases were not found or not detected at all. Brown rust occurred mainly on the flag leaf and sub-flag leaf. It appeared relatively late in the growing season, usually as late as June. The highest infestations were found in the same years as the drought occurred. As a result, the set of varieties tested in 2018 and those from 2022 were particularly susceptible to this pathogen. In the period 2019–2022, the varieties tested were very resistant, as they were in the dry and hot year of 2019. Many varieties had an average low infestation of brown rust in the range of 1–4%: Artist, RGT Kilimanjaro, Plejada, KWS Spencer, Bonanza, Hybery, Comandor, Medalistka, Delawar, Hondia, Owacja, Ostka GG, and RGT Bilanz. Unfortunately, these were not the highest grain yielding cultivars.

Ongoing research indicates that variety selection should also take into consideration resistance to brown rust, as this was usually linked to Septoria resistance and overall plant health. It should be noted that overall plant infestation was not negatively correlated with yield and thousand grain weight. Plant density was negatively correlated with infestation (r = −0.41, *p* < 0.001), which may suggest that fungal diseases reduce plant survival during winter. The level of brown rust infestation was also negatively correlated with plant density (−0.35, *p* < 0.001), but positively correlated with mass of grains (r = 0.20, *p* = 0.0007). No correlation with grain yield was found. The high infestation of brown rust is confirmed by the fact that resistance to this disease has not been a priority in winter wheat breeding process [22]. Furthermore, in our experiment, brown rust did not cause any known yield losses, as it appeared relatively late and the highest severity was caused by high temperatures. This may have been due to the fact that, although the rust occupied a large portion of the assimilative part of the leaf, it did not completely disrupt photosynthesis, in contrast to other diseases such as Septoria and tan spot. An interesting note may be the fact that plants heavily infected by brown rust had a lower proportion of Septoria and fusariosis, which paradoxically may have a positive effect on yield. However, there is no evidence for this relationship either in the results obtained or in the literature reviewed.

Septoria had the highest infestation when the proportion of brown rust decreased. The average Septoria infestation was only 8.9%. Varieties in the 2–4% range were Plejada, Hybery, Artist, RGT Kilimanjaro, Argument, LG Keramik, Euforia, Delawar, RGT Bilanz,

Comandor, Bonanza, KWS Spencer, and Symetria. Septoria was most intense when accompanied by one to three other diseases (Table 10). The occurrence of Septoria symptoms affected the overall leaf infestation (r = 0.49, *p* < 0.001). Septoria was the second most severe disease found on the flag leaf in the last five years. Symptoms of Septoria were observed to appear as early as the stalk shooting stage, in late May/early June, making it the main plant-wide disease. Later, depending on the weather pattern, it usually gave space to brown rust. Septoria can cause leaf withering on the lower level of the plant. On the uppermost leaf it is only a problem for the most susceptible varieties. Septoria infection was negatively correlated with grain yield (r = −0.25, *p* < 0.001) and plant density (r = −0.32, *p* < 0.001). Despite the very low average Septoria infestation, its problem should not be ignored. The results from the end of the growing season may reflect poorly on what was happening earlier. Septoria attacks plants in autumn and in early spring, so actual losses may not have been reflected in this monitoring.

**Table 10.** Fungal disease infestation in relation to the number of diseases found on a single leaf.

| Number of Identified Diseases | Total Infestation [%] | Septoria [%] | Tan Spot [%] | Brown Rust [%] | Yellow Rust [%] | Powdery Mildew [%] | Fusariosis and Other [%] |
|---|---|---|---|---|---|---|---|
| 1 | 37.75 | 3.88 | 0.46 | 32.36 | 0.95 | 0.10 | - |
| 2 | 35.73 | 8.72 | 0.40 | 23.93 | 2.23 | 0.46 | - |
| 3 | 28.34 | 9.25 | 1.96 | 13.82 | 1.90 | 1.40 | 0.01 |
| 4 | 22.87 | 8.46 | 5.52 | 5.80 | 1.03 | 1.07 | 0.99 |
| 5 | 10.96 | 2.71 | 3.09 | 2.04 | 1.08 | 1.04 | 1.00 |
| >5 | 7.19 | 1.49 | 1.57 | 1.12 | 1.00 | 1.02 | 1.00 |

The same was true for tan spot, which occurred mainly where the dominance of other diseases was low. The highest infestation of this disease was found on leaves where four and five different diseases were present, and in years with optimal weather. It occurred mainly not on the flag leaf, but on the lower plant levels, colonizing them relatively late in the growing season. The disease was more prevalent in plants with delayed development compared to the canopy. The prolonged vegetation of plants under favorable conditions influenced the higher infestation. It was also observed that there were more symptoms on the neighboring spring wheat than on the winter wheat tested [23]. For most of the varieties, tan spot was not a particular threat. The level of infestation was correlated with plant density (r = 30, *p* < 0.001) but not with the yield obtained. The average cover of tan spot infestation was only 2%, with the most resistant varieties being Argument, SY Dubai, Almari, LG Keramik, RGT Provision, Impresja, Kariatyda, SY Yukon, Fidelius, Arktis, Linus, and KWS Universum.

Yellow rust was rare and did not usually reach high leaf coverage values. It was most often found on leaves with two to three other diseases in association with brown rust (Table 10). The presence of yellow rust was thus positively correlated with overall plant infestation (r = 0.17, *p* < 0.001), even though it did not reach significant values on its own. With a very heavy infestation of brown rust, other rust species were difficult to distinguish. Yellow rust was not significantly correlated with yield obtained. In the future, yellow rust may become more important as changing climatic conditions begin to favor this species [22]. Other species of wheat rusts, especially leaf rust *Puccinia triticina* Erikss. can cause 50% losses in wheat production worldwide, and it is much more serious threat than yellow and brown rust [33]. Powdery mildew occurred at a relatively constant intensity of around 0.5–1%; however, it was poorly detected with high densities of other diseases. Powdery mildew also often produced nonspecific symptoms that could go unrecognized. Winter wheat yield results were independent of the presence of this disease. This does not coincide with the observations of other researchers, as powdery mildew was reported as

one of the most yield-limiting diseases. On the other hand, biological progress in winter wheat breeding was largely geared towards resistance to this pathogen [3,22].

Other diseases were identified mainly on leaves with low overall infestation (−0.46, $p < 0.001$). These mainly included leaf fusariosis, but also numerous unrecognized symptoms of other pathogens such as *Rhynchosporium* sp., *Ascochyta* sp., and possibly bacterial and viral infections. Fusarium usually causes ear and stem base diseases; however, we found symptoms on leaves, which could or could not be linked to the subsequent occurrence on the ears. Leaf fusariosis is very rare foliar disease; even infected leaves usually show no symptoms. We managed to identify the characteristic extensive, light-colored patches with a dark bordering on flag leaves of almost all varieties in 2020 and on some in 2021. In these particular years, the dominant diseases had a very low appearance, allowing Fusarium to gain a significant share of the infestation in some varieties (Figure 4). Those cultivars of winter wheat were also ranked as the most resilient ones, but on the other hand, they did not achieve maximal yield or best TGM. This fact may be attributed to the increased Fusarium presence; however, ears were not assessed in this research. Other authors using inoculation and DNA extraction successfully identified numerous Fusarium species, both in the ears and in the leaves of wheat [34,35]. Except for fusariosis symptoms, which account for about 1% of total infestation, other diseases were identified rather occasionally. Only in 2020 and 2021 did such symptoms occur at an intensity approaching 1%. Despite the low incidence, these cereal diseases could have had a significant impact on the yield obtained. A negative correlation was found between leaf fusariosis infection and crop yield ($r = −0.21$, $p = 0.0003$) and grain mass ($r = −0.44$, $p < 0.001$). The correlation with plant density was positive ($r = 0.44$, $p < 0.001$), which may mean that fusariosis occurs more frequently at high plant densities.

The results provide an insight into the relationship between the local climate and the occurrence of fungal diseases. It was also possible to identify relationships between different disease species ascending on wheat leaves. The selected varieties can be applied in areas with a similar climate in the area of Central European Lowlands: Jutland Peninsula, North Sea Coast, South Baltic Coast, South Baltic Lake District, Saxon–Lusatian Lowlands, Central European Lowlands, and Brabant–Westphalian Lowlands. On the above-mentioned regions similar weather conditions occur; however, it has to be highlighted that the studied location is on the eastern border of Central European Lowland, meaning the continental climate may affect it more than oceanic. At the western part of this geographical area, higher precipitation and lower summer temperatures may occur; however, soil types and soil quality remain mostly comparable. Therefore, the same varieties of winter wheat can be applied in whole this area, and the effect such as that in 2019–2021 would be expected for organic farming. It should be encouraged to test those varieties in different regions for their performance in organic farming also using allowed plant protection products.

## 4. Conclusions

- The course of weather determined the incidence of fungal diseases. The highest infestation occurred in years with low precipitation sums in years 2018, 2019, and 2022. Drought preceding plant maturation caused plant weakening and increased susceptibility, particularly to brown rust. The most favorable conditions for diseases occurred when the first half of the year was dry and warm and the second half was wet and cold.
- The most important fungal disease of organic winter wheat was brown rust, which determined the total infestation and the occurrence of other diseases. The second most occurring disease was Septoria leaf spot. Diseases such as tan spot, yellow rust, powdery mildew, and fusariosis had a lesser impact on crops.
- Leaf Septoria was associated with the highest yield losses. High-yielding varieties are often susceptible to brown rust.
- At least 10 different varieties with similar results can be classified as resistant to fungal diseases. Such a broad recommendation will maintain diversity in agriculture

and this will allow for the possible selection of varieties and mixtures, depending on the purpose of the crop and the location. The result can be applied over a wide geographical area.

**Author Contributions:** Conceptualization, K.J. and B.F.-S.; methodology, K.J., B.F.-S., P.R., and T.J; validation, P.R. and B.F-S; formal analysis, P.R.; investigation, P.R. and T.J.; resources, K.J.; data curation, P.R. and T.J.; writing—original draft preparation, P.R.; writing—review and editing, B.F.-S.; visualization, P.R.; supervision, K.J.; project administration, K.J.; funding acquisition, K.J. All authors have read and agreed to the published version of the manuscript.

**Funding:** The study was carried out as part of task 4.2 "Assessment of suitability of varieties of spring and winter cereals and legumes for organic farming system" of the budget subsidy for the implementation of the tasks of the Ministry of Agriculture and Rural Development in 2023.

**Institutional Review Board Statement:** Not applicable.

**Informed Consent Statement:** Not applicable.

**Data Availability Statement:** Data are available on request.

**Acknowledgments:** We would like to thank Polish Ministry of Agriculture and Rural Development for supporting our research efforts. We would like to address our gratitude to the staff of the Agricultural Experimental Station "Kępa" in Puławy, for taking care of the research fields. Finally, we thank the Reviewers contributing to the improvement of our study.

**Conflicts of Interest:** The authors declare no conflict of interest.

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
