# Peer review of "Assessment of Resistance of Different Varieties of Winter Wheat to Leaf Fungal Diseases in Organic Farming"

_agriculture, doi:10.3390/agriculture13040875_

Round 1
Reviewer 1 Report
I found the manuscript very interesting, however i think some spelling mistakes were made:
line 22: write Fusarium
line 33: correct the comma
line 50: however?, delete it, and re write the next sentence
line 90: correct spaces
line 105: iven?
line 129: is?
along the all manuscript write: 8.9 °C in stead of 8.9°C
line 134: 6.3 ha, divided into... (delete . It is)
line 255: It suggests conclusion???, re write
line 270: No less?
line 377: dubble space
in line 495 you write "Table 6", but in line 564 "tab 7", they must be written equally.
line 558: ".. there ir no evidence in the for this..." re write this sentence
line 563: dubble space
line 574: "as soon as", re write the sentence
line 580: delete the comma
line 582: there were
line 583: dubble space
line 585: the average of.., re write the sentence
line 597: detected?, re write the sentence
Author Response
Response to Reviewer 1 Comments
I found the manuscript very interesting, however i think some spelling mistakes were made:
Dear Reviewer 1, We would like to thank you for your time and effort put into the improvement of our manuscript. We have attempted to respond to all comments. Please note that we are referring to specific lines in the document with visible "Track Changes" and not to the "clean version" of the document.
line 22: write Fusarium
Corrected, line 22
line 33: correct the comma
Corrected, line 33
line 50: however?, delete it, and rewrite the next sentence
Deleted, line 50. Sentence was rewritten, lines 50-53
line 90: correct spaces
Corrected, line 90
line 105: iven?
Corrected, we mean ‘even’, line 105
line 129: is?
Corrected, we mean ‘was’, line 129
along the all manuscript write: 8.9 °C in stead of 8.9°C
Throughout the manuscript, values and units were separated by spaces, as indi-cated.
line 134: 6.3 ha, divided into... (delete . It is)
It has been corrected as suggested, line 134
line 255: It suggests conclusion???, re write
The sentence was rewritten, lines 257-259
line 270: No less?
Corrected, line 275
line 377: dubble space
Corrected, line 342
in line 495 you write "Table 6", but in line 564 "tab 7", they must be written equally.
Table and figure citations have been standardized throughout the manuscript (Table 1, Figure 1)
line 558: ".. there ir no evidence in the for this..." re write this sentence
Sentence was corrected, lines 615-616
line 563: dubble space
Corrected, line 621
line 574: "as soon as", re write the sentence
The sentence was rewritten, lines 631-632
line 580: delete the comma
coma was removed, line 676
line 582: there were
It was corrected, line 678
line 583: dubble space
Space was removed, line 679
line 585: the average of.., re write the sentence
It was corrected, line 681
line 597: detected?, re write the sentence
We mean ‘detected’ line 696
Reviewer 2 Report
The manuscript entitled "Assessment of resistance of different varieties of winter wheat to leaf fungal diseases in organic farming " is designed to assess the effect of local climatic conditions on the incidence of major fungal diseases and the identification of a possible selection of fungal-resistant winter wheat varieties for organic cultivation. There are a few minor changes required before the acceptance of the manuscript.
1. Specify the number of varieties in the abstract section on line 10.
2. Include the correlation matrix in a figurative form that will help readers.
3. Provide a table including the detail of the yield of all varieties.
4. In the discussion section, add some recent studies that favor your results.
Author Response
Response to Reviewer 2 Comments
Dear Reviewer 2, thank you for your valuable recommendations to improve our work. We have done our best to make all the required changes. Please note that we are referring to specific lines in the document with visible "Track Changes" and not to the "clean version" of the document.
- Specify the number of varieties in the abstract section on line 10.
It was specified that 41 varieties of winter wheat were tested. Line 14.
- Include the correlation matrix in a figurative form that will help readers.
Correlation matrices have been added. Table 2 - line 282, Table 4 - line 363 and Table 7 - line 445.
- Provide a table including the detail of the yield of all varieties.
Information on yield and thousand grain weight has been added to Table 9. Line 543.
- In the discussion section, add some recent studies that favor your results.
Eight additional studies were analysed and used in the discussion. Lines: 274, 334, 377, 563-576, 694, 713, and 822-844 in references.
Reviewer 3 Report
Dear Authors,
The study was carried out in Poland, in the narrow area of Osiny, in 2018-2022. In this region, 12 varieties in 2018, in later years the number 16 varieties were planted, and organic agriculture discipline was applied in this region.
The authors evaluated the varieties for their resistance against brown rust and Septoria disease, using scales for the diseases. The experiments were performed well and the results seem good. The authored tried to address every aspects of weather conditions (temperature, precipitations and climatic water balance) on plant health.
In the Result and Discussion section,
However, the study failed to provide enough discussion the results with previous studies. The authors could have discussed the results of the study by discussing with more study from different regions of the country or world provided some useful insights on disease management.
I strongly suggest that each criterion (temperature, precipitations, and climatic water balance) be adequately discussed separately.
In this way, the climate data is so crowded in the text that it is difficult to understand the results.
I think the results of the study would be more informative if the result and discussion sections were separated.
· Line 142: How to evaluate diseases other than Septoria and brown rust disease?
· What is the relevant scale of tan spot disease, it needs to be added. How was the data obtained?
· Although fusariosis is not a leaf disease, why was it included in the study? How was the data obtained?
You can find some point corrections in the text in the pdf file I sent.

Author Response
Response to Reviewer 3 Comments
The authors evaluated the varieties for their resistance against brown rust and Septoria disease, using scales for the diseases. The experiments were performed well and the results seem good. The authored tried to address every aspects of weather conditions (temperature, precipitations and climatic water balance) on plant health.
Dear Reviewer 3,
Thank you for your favourable assessment of our work. We will try to address all comments. Please note that we are referring to specific lines in the document with visible "Track Changes" and not to the "clean version" of the document.
In the Result and Discussion section,
However, the study failed to provide enough discussion the results with previous studies. The authors could have discussed the results of the study by discussing with more study from different regions of the country or world provided some useful insights on disease management.
We have made efforts to improve the quality of discussions. We have taken into consideration recent studies by other authors. Unfortunately, much of the work on this topic and in this geographical region was done by ourselves or by co-authors. At the same time, the journal's policy obliges us to avoid self-citation. We hope that the studies used are sufficient. Please see the lines: 274, 334, 377, 563-576, 694, and 713.
I strongly suggest that each criterion (temperature, precipitations, and climatic water balance) be adequately discussed separately.
The recommended changes have been applied. For this purpose, 3 additional sections were dedicated in the results:
3.1 Effect of the air temperature on plant health and severity of fungal deses (line 198)
3.2 Influence of precipitation on the incidence of fungal diseases (line 285)
3.3 Climatic water balance and plant helth (line 383)
In this way, the climate data is so crowded in the text that it is difficult to understand the results.
Correlation matrices have been added to enhance readability. Unnecessary references to previous sections have been removed and the individual discussion for temperature, precipitation and climatic water balance has been elaborated.
I think the results of the study would be more informative if the result and discussion sections were separated.
We must agree with this comment, however, the form of combining discussion and results is allowed in this journal. It seemed to us that an instant discussion of the results in the proximity of figures and tables would enhance the coherence of the paper. We kindly ask you and editors to accept the proposed form.
- Line 142: How to evaluate diseases other than Septoria and brown rust disease?
This is a valid point about methodology, which we did not specify. All blotch-leaving diseases are assessed on the same scale as Septoria (Figure 2). Similarly, all rusts are rated on the same scale (Figure 1). The methodology has been revised (lines: 149-152).
- What is the relevant scale of tan spot disease, it needs to be added. How was the data obtained?
Tan spot was assessed using the same method as Septoria. A significant infestation can be considered to be 10-50%. These values are obtained on some spring wheat varieties at this location (data not published).
- Although fusariosis is not a leaf disease, why was it included in the study? How was the data obtained?
This is a very interesting point that we need to address further. Indeed, Fusarium mainly causes diseases of the ear and the base of the stalk, as well as seedling rot. The symptoms observed on flag leaves are very rare and often confused with other diseases. They are quite characteristic, forming extensive light spots with a dark or black border. In 2020, we identified numerous occurrences of this disease on winter wheat after rains. It also occurred in small numbers in 2021. In the other years, it was not found at all. Until completing the literature review, we did not realise how rarely this disease is found on leaves. Please see lines 703-723. Fusarium was assessed on the same scale as Septoria (lines: 149-152).
You can find some point corrections in the text in the pdf file I sent.
Thank you for the PDF file. Corrections have been made in the lines: 33, 36, 93, 99, 105, and 220. References to tables have been standardised throughout the text.
Round 2
Reviewer 3 Report
Dear Editor,
I no longer have serious criticisms regarding methodology, results and interpretation of the results. The contents of the revised manuscript would be of considerable interest to readers in Agriculture.
Sincerely